# Assessment of Brain-Derived Neurotrophic Factor (BDNF) Concentration in Children with Idiopathic Nephrotic Syndrome

**DOI:** 10.3390/ijms232012312

**Published:** 2022-10-14

**Authors:** Andrzej Badeński, Marta Badeńska, Elżbieta Świętochowska, Agnieszka Didyk, Aurelia Morawiec-Knysak, Maria Szczepańska

**Affiliations:** 1Department of Pediatrics, Faculty of Medical Sciences in Zabrze, Medical University of Silesia, 40-055 Katowice, Poland; 2Department of Medical and Molecular Biology, Faculty of Medical Sciences in Zabrze, Medical University of Silesia, 40-055 Katowice, Poland; 3Department of Pediatric Nephrology with Dialysis Division for Children, Independent Public Clinical Hospital No. 1, 41-800 Zabrze, Poland

**Keywords:** idiopathic nephrotic syndrome, BDNF, children, marker

## Abstract

Idiopathic nephrotic syndrome (INS) is a chronic disease affecting children in early childhood. It is characterized by proteinuria, hypoalbuminemia, edema and hyperlipidemia. To date, the diagnosis is usually established at an advanced stage of proteinuria. Therefore, new methods of early INS detection are desired. This study was designed to assess brain-derived neurotrophic factor (BDNF) as a potential marker in the early diagnosis of INS. The study group included patients with a diagnosis of idiopathic nephrotic syndrome (*n* = 30) hospitalized in Clinical Hospital No. 1 in Zabrze, from December 2019 to December 2021. Our study shows that serum BDNF concentration decreased and urine BDNF concentration increased in a group of patients with INS, compared with healthy controls. Such outcomes might be related to loss of the BDNF contribution in podocyte structure maintenance. Moreover, we anticipate the role of BDNF in urine protein concentration increase, which could be used as a direct predictor of urine protein fluctuations in clinical practice. Moreover, the ROC curve has also shown that serum BDNF and urine BDNF levels might be useful as an INS marker.

## 1. Introduction

### 1.1. The Idiopathic Nephrotic Syndrome

Idiopathic nephrotic syndrome (INS) is a glomerular disease, most frequently with an early childhood onset. Pathophysiology of INS is based on a podocyte damage, leading to disturbances in the glomerular filtration barrier functioning [1]. INS is defined by nephrotic-range proteinuria (≥40 mg/m^2^/hour or urine protein/creatinine ratio ≥200 mg/g or 3 + protein on urine dipstick), hypoalbuminemia (<25 g/L) and edema [2,3].

In the majority of patients, the clinical course is characterized by frequent relapses, often related to infectious events. Patients are classified by their response to steroid therapy. Most INS children require long-term steroid and/or immunosuppressive therapy aiming to control disease activity. Such control is usually achieved within 4 weeks of steroid therapy (steroid-sensitive NS [SSNS]). Unfortunately, 50% of patients are likely to become steroid-dependent or present with frequent relapses of the disease [4,5]. Studies have proven that the more relapses a patient develops, and whether there is a need for non-steroidal immunosuppressants, the more active the disease will be in early adulthood [6,7,8,9]. For SSNS patients, after 10 years, the risk of progression to chronic kidney disease is quite low, at about 5% [10], while children with steroid-resistant INS are much more likely to develop end-stage renal disease, mainly those who do not achieve remission [11]. 

Nevertheless, the determinants of a long-term prognosis for INS patients should be investigated more [1].

### 1.2. The Similarity of Neuronal Cells to Podocytes

Podocytes maintain the glomerular filtration barrier through their unique cellular structure, which includes major processes and foot processes covering the glomerular basement membrane (GBM) [12,13]. 

A typical neuron comprises the cell body (perikaryon) and its projections: the axon and dendrites. Neurons contact each other through synapses to form neural networks.

Podocytes, similarly to neurons, show a complex system of branches consisting of the major processes, which are further divided into foot extensions. The branches of podocytes alternate with those that depart from neighboring cells, also forming a tight network that completely surrounds the basal membrane of the glomeruli [14].

These two cell types share a dynamic actin-based cytoskeleton architecture. It is worth mentioning that actin is mainly concentrated in specialized, thin branches—in the foot processes in podocyte cells, and in dendritic extensions in neurons. Such structures enable them to keep their desired shape and function [15,16].

In neuronal cells, actin remodeling is the driving force for the formation, shaping, and stabilization of dendritic projections. Studies on neuronal and psychiatric disorders, such as Alzheimer’s disease or schizophrenia, have defined the underlying pathology as a damage of these processes, which, in turn, results in a reduced number and altered morphology of dendritic projections [17]. Similarly, the dynamics of podocyte actin serves to maintain the correct shape and arrangement of the foot extensions along the glomerular basement membrane. In patients with focal segmental glomerulosclerosis (FSGS), minimal change nephropathy and diabetic nephropathy (DN), architectural changes in the glomeruli are observed in the form of reduction of foot processes [18,19].

### 1.3. Brain-Derived Neurotrophic Factor (BDNF)

BDNF is a pleiotropic neurotrophin that binds to the tropomyosin-related kinase B (TrkB) receptor and plays a key role in the maturation, survival and activity of neurons. TrkB is also produced in the renal glomeruli, collecting tubules, and the glomerular apparatus [20,21,22,23]. In addition, BDNF has been shown to have TrkB-dependent trophic activity on podocyte foot cells, resulting in actin polymerization. In vitro exposure to BDNF results in an increase in length and number of podocyte branches [14]. Similar changes were repeatedly observed in neuronal cells, where the application of BDNF increased axon length and the number of dendritic spikes and synapses. Importantly, BDNF is effective in repairing podocyte damage in vitro, opening up a potential new perspective on treating podocyte disorders [24,25]. BDNF levels in serum and urine have not yet been determined in a pediatric population with INS.

### 1.4. Main Aims of the Study

INS is a chronic, progressive disease affecting children in early childhood. Up to this point, the diagnosis has usually been established at an advanced stage of proteinuria. Therefore, new methods of early INS detection are desired. This study was designed to assess BDNF as a potential marker in the early diagnosis of INS, as well as a predictor of proteinuria, also among patients with relapse of the disease.

## 2. Results

### 2.1. Descriptive Analysis and Comparison—INS Patients and Control Group

All 74 patients met the inclusion criteria, among which 30 represented the study group and 44 represented the control group. The study group included 14 girls (46.6%) and 16 boys (53.4%), with a mean age of 7.67 ± 4.14 years. The control group consisted of 18 girls (40.9%) and 26 boys (59.1%), with a mean age of 7.75 ± 4.10 years. Among children with INS mean urine protein level was 4.45 ± 8.78 g/L; however, serum creatinine levels were within normal range for all patients with mean creatinine level at 34.00 ± 14.60 μmol/L. 

Seven individuals (23.3%) from the study group were admitted to the clinic with an initial manifestation of INS, whereas the remaining twenty-three patients (76.7%) were hospitalized due to a relapse. 

Cyclosporine therapy was conducted in 10 cases (33.3%), with only 1 patient from the group receiving cyclophosphamide, whereas the rest of the group (19 individuals) were treated with steroids only. 

Patients from the control group were not diagnosed with any kidney disorders, neither congenital nor acquired, whereas the study group only included patients with INS.

Table 1 presents the full characteristics of the studied groups of children with INS and the control group.

### 2.2. BDNF Results

To investigate whether BDNF concentration was dependent on the INS manifestation, we used Student’s *t*-test for independent samples of both urine and serum. Significantly higher concentrations of BDNF in urine (*p* = 3.9 × 10^−10^) and significantly lower serum concentrations (*p* = 2.6 × 10^−8^) were observed in the study group compared with the control group (Figure 1). 

Additionally, we determined the serum BDNF to serum creatinine ratio for both the study and control groups. Among patients with INS, the mean BDNF/creatinine ratio was 0.3, whereas in the study group, it reached 0.02 (Table 2).

To identify possible correlations between serum and urine BDNF concentrations, we applied the Pearson correlation coefficient separately for both groups. There was a significant positive correlation in the control group (R = 0.84, *p* < 0.0001); however, in the study group, the correlation was not statistically significant (Figure 2).

We assessed the possible usefulness of BDNF as an INS marker using the ROC curve. The suggested cut-off point for serum BDNF was established at 0.9 pg/mL, with the AUC at 86.5%. Additionally, the urine BDNF cut-off point reached 1.0 pg/mL with AUC at 92.0% (Figure 3).

Furthermore, there were no significant differences in serum or urine BDNF among INS patients treated with cyclosporine and those on other medication (Figure 4). 

The general linear model showed a significant negative correlation between the concentration of BDNF in the blood serum and the intensity of proteinuria. On its basis, it was established that a 1 unit decrease in serum BDNF concentration increases the amount of protein in urine by 26 units. It proves more precisely that the concentration of BDNF in the serum is directly related to the intensity of proteinuria (Table 3). Other significant correlations were not detected.

## 3. Discussion

### 3.1. BDNF as an INS Marker

As mentioned before, diagnosis of INS is established based on nephrotic-range proteinuria, hypoalbuminemia and edema. Such characteristics are a consequence of the leakage of protein from the blood into the urine through damaged glomeruli. Therefore, we assume that the discovery of new markers for the disease is crucial.

In our study, we assessed BDNF as a potential marker for podocyte damage in serum and urine. Significantly lower serum concentrations of BDNF in the study group, in addition to higher urine levels, might suggest that the concentration of serum BDNF decreases in cases of kidney damage. Regarding the chosen treatment for INS, our study revealed that there is no substantial influence on BDNF concentrations during cyclosporine therapy.

Moreover, the prediction of the sensitivity and specificity of BDNF as a marker showed a promising outcome. The ROC models presented high AUC (85.6% for serum concentration and 92% for urine concentration), which means that it had a good measure of separability. However, the study group size in this study seemed to be a limitation for such conclusions; therefore, further assessments on larger groups of patients should be conducted.

Linear regression analysis proved that the concentration of BDNF in the serum is directly related to the intensity of proteinuria. We believe that BDNF may also serve as a predictor of urinary protein levels, especially among patients in remission, serving as an early detector of a relapse. Nevertheless, further research in this field is still required.

### 3.2. BDNF as a Potential Therapy for Damaged Podocytes

Actin is responsible for the shape and function of podocytes as the main cytoskeletal component of podocytes’ processes. During podocyte damage, many changes occur, such as a loss of stress fibers, cell rounding and shortening of branches in vitro, as well as clogging of the major process and the transformation of microvilli in vivo. These changes are related to the evolution of actin dynamics [26].

The dynamics and reorganization of actin filaments are regulated in time and space by numerous actin-binding proteins and upstream signaling molecules that collectively control the folding and unfolding of actin filaments [27].

Proteins from the actin depolymerization factor (ADF)/cophilin family play a key role in the reorganization of actin filaments. They stimulate the depolymerization of actin filaments [28]. Cophilin-1 is involved in the homeostasis of the podocyte foot process [29].

The phosphorylation of cofilin-1 leads to its inactivation, which, in turn, is mainly regulated by Limk1 kinase [30]. In dendritic spines, Limk1 is kept under the direct control of miRNA-132 and miRNA-134, which further confirms the similarities between the dynamics of the dendritic spine and the podocyte process [14] (Figure 5).

Li et al. posited the hypothesis that, in damaged podocytes, BDNF might contribute towards prolonging the podocyte processes, which contrasts with the flattening induced under nephrotic conditions both in vitro and in vivo, opening up a potential treatment option for damaged podocytes, especially in focal segmental glomerulosclerosis (FSGS) [14]. 

On the other hand, Hahn et al. performed one study which suggests a possible correlation between IgA nephropathy (IgAN) and single nucleotide polymorphisms (SNPs) of genes encoding BDNF, including BDNF rs11030104 (intron), BDNF rs7103411 (intron), BDNF rs7103873 (intron), and BDNF rs6484320 (intron). Such outcomes led to a conclusion that the abovementioned SNPs might contribute to a greater susceptibility for IgAN and the progression of disease, because they could affect the expression of the protein and its function [31]. Therefore, we might assume that genetic polymorphisms of BDNF might have various impacts on podocyte structure; broader research in this field is required.

The promising actin-trophic activity of BDNF encouraged us to conduct other several studies, where this potential was exploited as a treatment in several neurological diseases. Positive results were obtained in experimental models of neurological diseases [32,33,34]. However, in clinical trials, crossing the human blood–brain barrier by BDNF appeared to be a significant obstacle [35].

TrkB is also expressed by several types of neuronal, epithelial, and connective neoplastic cells [36]; therefore, it can be assumed that when trying to use BDNF as a drug in diseases with podocyte damage, it will be necessary to deliver BDNF targeted at podocytes, to increase its effectiveness and specificity.

## 4. Materials and Methods

### 4.1. Studied Groups—Characteristics, Laboratory Outcome and Anthropometric Measures

The study group included patients aged 2 to 17 years with a diagnosis of idiopathic nephrotic syndrome (*n* = 30) hospitalized in the Department of Pediatric Nephrology with the Subdivision of Dialysis at the Clinical Hospital No. 1 in Zabrze, Medical University of Silesia in Katowice in the period from December 2019 to December 2021.

Inclusion criteria for this group were as follows: confirmed diagnosis of INS, patient newly diagnosed or in relapse of underlying disease, children older than 3 months (to exclude congenital NS) and less than 18 years of age. The exclusion criteria comprised: non-nephrotic proteinuria, congenital NS and NS secondary to metabolic, infectious, vascular, malignant, and cardiac diseases.

Idiopathic nephrotic syndrome was diagnosed on the basis of the Recommendations of Polish Society for Pediatric Nephrology dedicated to the management of the child with nephrotic syndrome from 2015 [37]. The medical history included information on the medications used, the recurrence of underlying disease, and the coexistence of comorbidities.

The control group (*n* = 44) consisted of patients hospitalized in the same period in the Department of Pediatric Nephrology with the Subdivision of Dialysis due to bedwetting or visiting the Department of Surgery of Child Developmental Defects and Traumatology of the Clinical Hospital No.1 in Zabrze, Medical University of Silesia in Katowice in order to undergo procedures as part of one-day surgery. These patients were not diagnosed with chronic diseases nor infectious diseases, and their kidney function was normal.

This research project was approved by the Ethics Committee of the Medical University of Silesia in Katowice (PCN/0022/KB1/133/19). Written informed consent was obtained from caregivers of all the children and, in cases where participants were older than 16 years, also from the child. 

The children underwent complete blood count and biochemical tests (serum urea, serum creatinine, serum uric acid, cholesterol level, level of triglycerides, total serum protein concentration, blood ionogram, blood gas test, and urinalysis). The estimated glomerular filtration rate (eGFR) was calculated using the classic Schwartz formula, taking into account the age-appropriate k-coefficient [mL/min/1.73 m^2^].

BDNF concentration in blood serum and urine was detected with ELISA test using the Cloud-Clone (USA)—Human BDNF kit, catalog number SEAO11Hu. The analytical procedure was in accordance with the technological instructions included in the kit by the manufacturer. Absorbance readings were taken on a SYNERGY/H1 reader (BioTek, Santa Clara, CA, USA) at a wavelength of 450 nm using a 620 nm reference wavelength. Elaboration of the results was performed using the Gen5 v 3.05 computer program (BioTek, Santa Clara, CA, USA). The sensitivity of the method reached 11.7 pg/mL. The precision of the method in the simultaneous series (imprecision) was 3.8%.

In all studied children, anthropometric parameters (weight, height) and blood pressure were determined. Weight was given in kilograms (with 0.01 kg precision), and height was measured with 0.01 cm precision using a standardized stadiometer. Based on such data for each patient, the body mass index (BMI) was calculated (using the equation: weight/height^2^ (kg/m^2^)). 

Weight, height, BMI and blood pressure were plotted for age and sex using the percentile charts as a result of the OLA and OLAF trials (estimated for polish children population) [38,39].

To compare blood pressure, body weight, BMI, and height between different groups of children, SDS values for systolic and diastolic blood pressure, body weight, BMI and height were calculated.

There were no missing data in the collected groups.

### 4.2. Statistical Analysis

Statistical analysis was performed using R Studio software. Descriptive statistics were presented using means with standard deviation and medians with quartile range. Normal distribution was assessed using the Shapiro–Wilk test. The assessment of the homogeneity of variance was performed using the Levene’s test. The Pearson correlation coefficient was used to assess the correlation between the parameters. In the comparative analysis between the parameters of the control group and the study group, the *t*-test for independent variables was used. The usefulness of the tested proteins as disease markers was assessed through the ROC curve using the bootstrap method to determine cutoff point and 95% confidence interval.

*p* values < 0.05 were considered statistically significant. For more precise assessments of the association between BDNF and proteinuria, general linear modeling was performed to detect potential correlations between the severity of proteinuria and the concentration of BDNF in serum, taking into account the age and sex of the patient. Autocorrelation was assessed using the Durbin–Watson test.

## 5. Conclusions

Our study shows that serum BDNF concentrations decrease in groups of patients with INS, compared with healthy controls. Such outcomes might be related to the loss of the BDNF contribution in podocyte structure maintenance. Moreover, according to our study outcome, we describe the possible use of BDNF as a predictor of urine protein concentration increase. These observations encourage further clinical studies on BDNF as an INS early diagnosis marker, as well as a potential remedy for damaged podocytes.

## Figures and Tables

**Figure 1 ijms-23-12312-f001:**
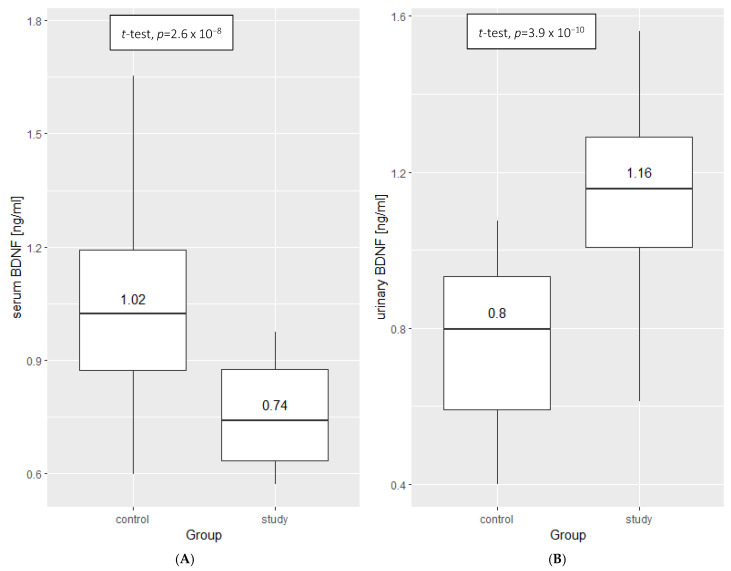
Comparison of BDNF concentration in serum (**A**) and urine (**B**) between the control group and the study group (Student’s *t*-test for independent samples). Additional data: mean ± standard deviation—(**A**): control: 1.02 ± 1.06, study: 0.74 ± 0.76; (**B**): control: 0.80 ± 0.77, study: 1.16 ± 1.14.

**Figure 2 ijms-23-12312-f002:**
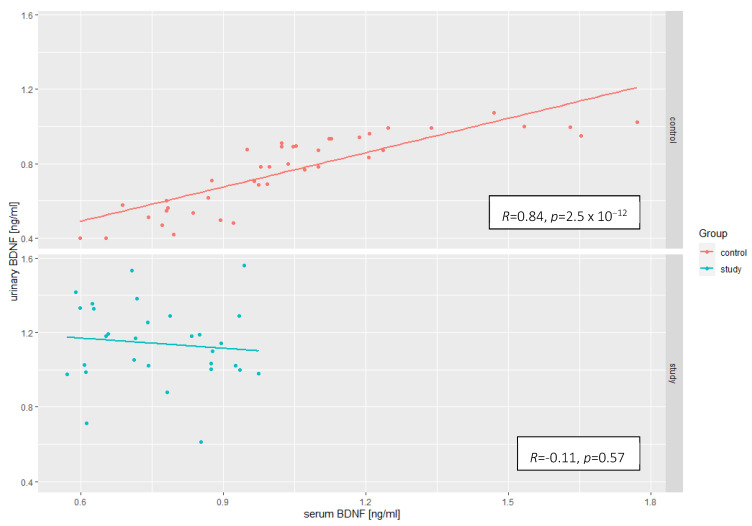
Correlations between the concentration of BDNF in the serum and urine, taking into account the division into the study group and the control group (R-Pearson’s correlation coefficient).

**Figure 3 ijms-23-12312-f003:**
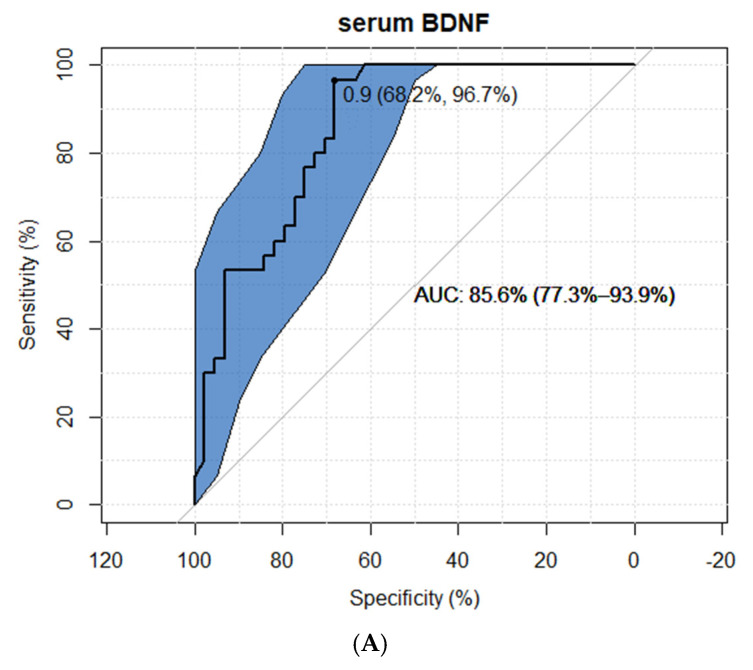
The assessment of usefulness of BDNF as an INS marker in serum (**A**) and urine (**B**) (ROC curve)—study group.

**Figure 4 ijms-23-12312-f004:**
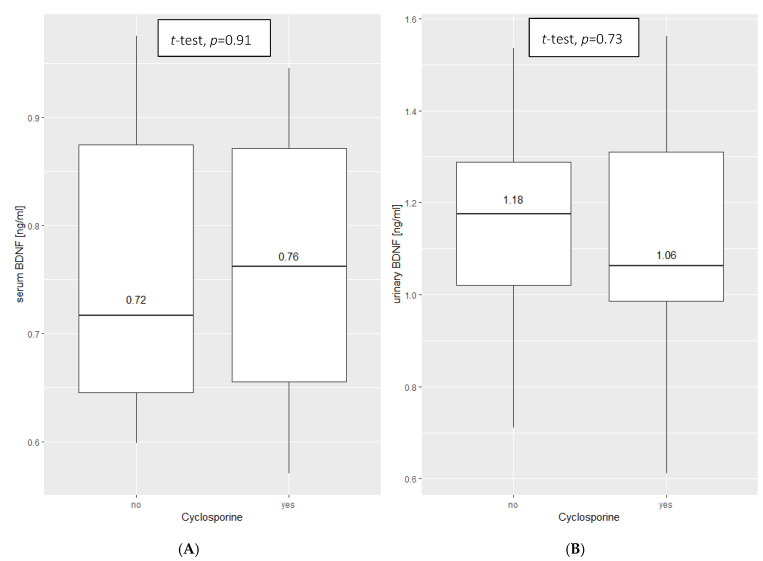
The comparison of BDNF concentration in the study group in serum (**A**) and urine (**B**) between the patients treated with cyclosporine (yes) and treated with other medicine (no), (Student’s *t*-test for independent samples).

**Figure 5 ijms-23-12312-f005:**
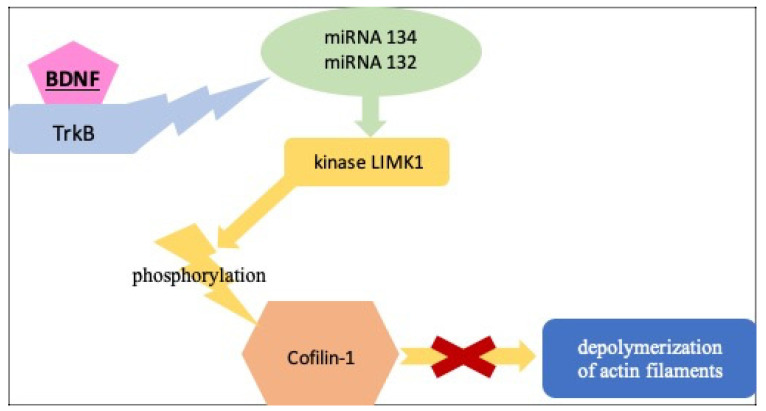
The role of BDNF in actin polymerization (based on [14,28,29,30]); BDNF—brain-derived neurotrophic factor, TrkB—Tropomyosin receptor kinase B, LIMK1—LIM Domain Kinase 1.

**Table 1 ijms-23-12312-t001:** Characteristics of the groups—selected anthropometric and laboratory parameters.

Parameter	Children with Idiopathic Nephrotic Syndrome (*n* = 30)			Control Group (*n* = 44)		
	Whole Group	Female	Male	Whole Group	Female	Male
Age (year)	7.50 ± 4.14	6.14 ± 3.35	9.00 ± 4.40	7.75 ± 4.10	8.33 ± 3.95	9.85 ± 4.17
	(2.00–17.00)	(2.00–13.00)	(2.00–17.00)	(2.00–17.50)	(2.00–17.00)	(4.50–17.50)
Height (cm)	125 ± 25.00	115.46 ± 19.24	134.41 ± 26.70	127 ± 24.90	127.50 ± 21.59	142.17 ± 25.60
	(92.00–179.00)	(92.00–158.00)	(92.00–179.00)	(82.00–197.00)	(82.00–170.00)	(109.00–197.00)
SDS for height	0.06 ± 1.36	−00.7 ± 1.20	0.08 ± 1.53	0.09 ± 1.10	−0.14 ± 1.08	0.35 ± 1.08
	(−2.80–2.29)	(−2.80–2.20)	(−2.80–2.29)	(1.53–3.00)	(−1.53–2.35)	(−1.34–3.00)
BW (kg)	29.80 ± 21.60	26.54 ± 13.55	43.26 ± 24.49	27.10 ± 18.80	30.35 ± 15.03	38.38 ± 20.68
	(12.10–88.90)	(12.10–59.80)	(14.50–88.90)	(9.70–87.50)	(9.70–71.00)	(15.00–87.50)
SDS for BW	1.15 ± 2.1	1.12 ± 1.58	1.75 ± 2.49	0.07 ± 1.11	0.08 ± 1.23	0.04 ± 1.04
	(−1.83–9.58)	(−1.83–3.80)	(−1.63–9.58)	(−2.34–2.12)	(−1.87–2.11)	(−2.34–2.12)
BMI (kg/m^2^)	18.90 ± 4.07	18.41 ± 3.20	20.92 ± 4.48	16.60 ± 3.45	17.52 ± 3.12	17.52 ± 3.72
	(13.50–30.90)	(13.82–23.98)	(13.50–30.90)	(12.40–26.50)	(13.90–24.60)	(12.40–26.50)
SDS for BMI	0.83 ± 2.10	0.07 ± 1.18	2.20 ± 2.26	−0.06 ± 1.18	0.25 ± 1.05	−0.25 ± 1.25
	(−2.18–8.17)	(−1.47–2.35)	(−2.18–8.17)	(−2.97–2.29)	(−1.38–2.29)	(−2.97–2.02)
SYS (mmHg)	115 ± 14.6	108.21 ± 14.87	117.88 ± 13.17	110.00 ± 11.30	107.17 ± 8.54	113.96 ± 12.25
	(81.00–152.00)	(81.00–138.00)	(102.00–152.00)	(85.00–134.00)	(89.00–122.00)	(85.00–134.00)
DIA (mmHg)	67.50 ± 10.90	64.71 ± 9.51	73.44 ± 10.66	70.00 ± 12.00	66.83 ± 11.45	69.96 ± 12.44
	(50.00–90.00)	(50.00–88.00)	(56.00–90.00)	(40.00–107.00)	(45.00–96.00)	(40.00–107.00)
MAP (mmHg)	82.50 ± 11.30	79.21 ± 9.97	88.25 ± 11.03	82.50 ± 10.70	80.28 ± 9.54	84.63 ± 11.23
	(67.00–111.00)	(67.00–100)	(71.33–111.00)	(58.70–116.00)	(62.33–102.33)	(58.67–115.70)
Urine protein (g/L)	4.45 ± 8.78	6.04 ± 6.15	10.28 ± 10.35	X	X	X
	(0.67–35.60)	(0.88–23.70)	(0.67–35.60)			

Data are presented as the mean ± standard deviation (minimum–maximum); BW: body weight; SDS: standard deviation score; BMI: body mass index; SYS: systolic arterial pressure; DIA: diastolic arterial pressure; MAP: mean arterial pressure; X: there was no protein in urine.

**Table 2 ijms-23-12312-t002:** Serum BDNF to serum creatinine ratio assessed in study and control group.

Parameter	Control Group (*n*= 44)	Study Group (*n* = 30)
Serum BDNF/serum creatinine	0.02 ± 0.01 (0.01–0.04)	0.03 ± 0.01 (0.01–0.06)

Data are presented as the mean ± standard deviation (minimum–maximum).

**Table 3 ijms-23-12312-t003:** A general linear model of serum BDNF and proteinuria.

	β	SE	*t*	*p*
(Intercept)	−18.86	9.73	−1.94	0.06
BDNF serum level	26.66	11.55	2.31	0.03
Gender	2.98	3.08	0.97	0.34
Age	0.69	0.38	1.82	0.08

Intercept = constant term—element of regression, not affected by independent variable; β—regression coefficient; SE—standard error; *t*-test statistics value for each parameter.

## Data Availability

The data presented in this study are available on request from the corresponding author. The data are not publicly available due to privacy issues.

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
