# Peer review of "Assessment of Brain-Derived Neurotrophic Factor (BDNF) Concentration in Children with Idiopathic Nephrotic Syndrome"

_ijms, 2022, doi:10.3390/ijms232012312_

Round 1
Reviewer 1 Report
Major Comments:
The manuscript is well written. However the authors should address the following concerns:
1. In the ‘Introduction’, the authors said ‘This study was designed to assess the BDNF as a potential marker in early diagnosis of INS’. But, in the ‘Conclusion’ they mentioned ‘according to our study outcome, we describe the possible use of BDNF as a predictor of urine protein concentration increase’. These statements have no parity. The major concern for this study is that during conclusion, the authors didn’t consider the reported data of the serum and urine levels of BDNF in chronic kidney disease and other type of nephropathies (Diabetic nephopathy, Unilateral Ureteral Obstruction nephropathy etc.) because in these pathologies, podocyte injury and proteinuria are very common. In fact, excess urinary BDNF has already been reported in diabetic nephropathy. Moreover, alteration in serum and urinary BDNF are widely demonstrated in brain disorders like Parkinson's disease, Alzheimer's disease, Huntington's disease, Schizophrenia, Epilepsy, Major Depressive Disorder, Psychiatric Disorders etc. All this important information is not even discussed in the manuscript. Thus, it is questionable that how excess urinary and low serum BDNF predicts specifically about the INS?
2. Authors should also compare total amount of excreted BDNF in 24 h between study group and Control because concentration of BDNF in urine depends on volume of excreted urine. Thus it can vary upon different excretion volume.
3. Did the authors confirm that the individuals of the control and study groups had no other type of kidney diseases? This is important because this could have affected the result and thus conclusion. Please mention in the manuscript.
4. Table 1: Sex wise data of the parameters for the control group are absent. Data fonts are not similar.
5. Table 2: Gender wise data are not shown. There are no units for the data. Standard deviations are very high.
6. Table 2 and Figure 1 represent the same data. Thus one of those is redundant.
7. Why the authors compare serum and urinary BDNF between the patients treated with cyclosporine and treated with other medicine to assess the usefulness of BDNF as an INS marker in serum and urine? Why the authors didn’t compare the data of the study (treatment) groups with the control group?
8. There is no mention of other medicine-treated group in Figure 4. Is it indicating the ‘no’ group?
9. Table 3: The data contain comma. Please correct. Why females are not considered?
10. Information related to INS is not sufficient in ‘Introduction’ as well as in ‘Discussion’ sections.
Reviewer 2 Report
Authors reported an assessment of brain-derived neurotropic factor (BDNF) concentration in children with idiopathic nephrotic syndrome from a hospital over a 2-year period. The finding is interesting and potentiality useful to provide insights into the monitoring of the condition, use of BDNF as an INS marker, as well as a potential remedy for damaged podocytes in INS. The write-up is straightforward but may be improved further before its acceptance for publication. Comments are given below for authors' consideration.
Abstract:
Suggest to include more info on (1) Methodology, esp. where the subjects were from/where the study was carried out (population, duration). (2) The results - As it is now, only one sentence talked about the result finding very briefly (line 19-20). This is a research article and abstract can be further improved by optimizing the word count limit.
Introduction: It is nice to have subsections (1.1, 1.2, 1.3..) but I am not sure if this is the usual format for the journal. This may be fine for a thesis but for a research article, the authors perhaps should condense the introduction and importantly highlight the knowledge gap, their hypothesis and objectives/aim.
Results: Well presented.
For Table 1 caption - instead of just "Characteristics..." can it be more specified? Which aspects of "characteristics" are these?
Table 2. Include the number of subjects.
For correlation - should R2 be used instead of R? Pls explain.
Discussion:
Discussion and inferences are generally supported. However, Figure 5 may be redundant as this is not directly a new finding from the study, but rather a general concept that has well been established. Furthermore, there is no caption provided for the figure.
Methods: Sound and robust.
Conclusion: Supported.
Reviewer 3 Report
The article entitled "Assessment of brain-derived neurotropic factor concentration in children with idiopathic nephrotic syndrom" is very well written and the research problem is novel and highly appreciated topic. Early detection of this disease very much need.
Introduction, aim of the study and results were well addressed and sufficiently informative. In figure 1 comparison of BDNF authors mentioned (a) and (b) for figures but in text they used (A) and (B), that must be corrected. However, this comparison study is very well conducted. Conclusion also very well written. The readability of this paper is high.
I strongly recommend this article to publish in IJMS journal.
Round 2
Reviewer 1 Report
The manuscript has been improved and can be accepted for publication.